# Pathophysiological Roles of Mucosal-Associated Invariant T Cells in the Context of Gut Microbiota-Liver Axis

**DOI:** 10.3390/microorganisms9020296

**Published:** 2021-02-01

**Authors:** Yoseph Asmelash Gebru, Mi Ran Choi, Ganesan Raja, Haripriya Gupta, Satya Priya Sharma, Ye Rin Choi, Hyeong Seop Kim, Sang Jun Yoon, Dong Joon Kim, Ki Tae Suk

**Affiliations:** Institute for Liver and Digestive Diseases, Hallym University, Chuncheon 24252, Korea; yoseph@hallym.ac.kr (Y.A.G.); choimi316@nate.com (M.R.C.); RG@hallym.ac.kr (G.R.); phr.haripriya13@gmail.com (H.G.); satyapriya83@gmail.com (S.P.S.); dpfls3020@gmail.com (Y.R.C.); kimhs2425@gmail.com (H.S.K.); ysjtlhuman@gmail.com (S.J.Y.); djkim@hallym.ac.kr (D.J.K.)

**Keywords:** MAIT cell, MR1, gut-microbiota, riboflavin metabolites, liver disease

## Abstract

Mucosal-associated invariant T (MAIT) cells are a subset of T lymphocytes expressing a semi-invariant T-cell receptor (TCR) present as TCR *V*α7.2-*J*α33 in humans and TCR *V*α19-*J*α33 in mice. They are activated by ligands produced during microbial biosynthesis of riboflavin that is presented by major histocompatibility complex class I-related (MR1) molecules on antigen-presenting cells. MAIT cells also possess interleukin (IL)-12 and IL-18 receptors and can be activated by the respective cytokines released from microbially stimulated antigen-presenting cells. Therefore, MAIT cells can be involved in bacterial and viral defenses and are a significant part of the human immune system. They are particularly abundant in the liver, an organ serving as the second firewall of gut microbes next to the intestinal barrier. Therefore, the immune functions of MAIT cells are greatly impacted by changes in the gut-microbiota and play important roles in the gut-liver pathogenesis axis. In this review, we discuss the nature and mechanisms of MAIT cell activation and their dynamics during different types of liver pathogenesis conditions. We also share our perspectives on important aspects that should be explored further to reveal the exact roles that MAIT cells play in liver pathogenesis in the context of the gut microbiota.

## 1. Introduction

T cells are lymphocytes that play a central role in the immune system and differ from other lymphocytes by the presence of a T-cell receptor (TCR) specific to a particular antigen on the cell surface. The receptor recognizes peptide and lipid-based antigens bound with antigen-presenting molecules, primarily major histocompatibility complex (MHC) class I and II molecules [1,2]. The MHC class I molecules mainly recognize and present peptide antigens by interacting with an antigen-specific *αβ* TCR, which is capable of undergoing random rearrangements of V (variable), D (diversity), and J (joining) gene segments enabling specific recognition of a broad range of antigens [3]. Therefore, T cells can express TCRs capable of recognizing diverse antigens from pathogens, tumors, and the external environment while maintaining memory and self-tolerance. Upon stimulation, T cells mediate immune responses to microbial infections, mainly by killing infected host cells and producing cytokines that regulate immunity and inflammation. Several studies have explored and described the importance and natures of various types of T cell subsets that comprise effector and memory immune functions in healthy as well as diseased individuals [4,5]. About two decades ago, Tilloy et al. discovered a specialized subset of *αβ* T cells expressing a dominant invariant TCR *α*-chain gene rearrangement in humans and mice [6]. These cells were later named “mucosal-associated invariant T (MAIT) cells” because of their higher abundance in the lamina propria of the mucosa at that time. This was demonstrated by the abundance of *V*α7.2-*Jα*33 transcripts in human gut biopsies and enrichment of *Vα*19-*J*α33 transcripts in mice lamina propria lymphocytes [7].

MAIT cells were first discovered as lymphocytes in blood expressing TCR that contains α-chains associated with a limited set of β segments later confirmed to be present as TCR *V*α7.2-*J*α33 in humans and TCR *V*α19-*J*α33 in mice [6,8]. The limited variability of the TCR in these cells intrigued researchers to investigate if the receptor is able to recognize only a limited diversity of antigens. In 2003, Treiner et al. confirmed that MAIT cells recognize antigens presented by the MHC class I- related (MR1) molecule mainly expressed by B cells [7]. It was then demonstrated that MR1 binds to intermediate metabolites produced during biosynthesis of riboflavin (vitamin B2) by certain microbes in the gut and interacts with MAIT cells [9]. The most studied members of these antigens are the enzymatically produced early intermediate 5-Amino-6-d-Ribitylaminouracil (5-A-RU) and its nonenzymatic products from its reactions with glyoxal, and methylglyoxal generating 5-(2-oxoethylideneamino)-6-d-ribitylaminouracil (5-OE-RU) and 5-(2-oxopropylideneamino)-6-d-ribitylaminouracil (5-OP-RU) respectively. The naturally occurring folic acid (Vitamin B9) photo-degradation product, 6-formylpterin (6-FP) is also captured by MR1. However, unlike the riboflavin intermediates, this does not result in the stimulation of MAIT cells [10]. Because viruses are not capable of synthesizing riboflavin, the response of MAIT cells is suggested to be only through a cytokine-dependent pathway which is different from the MR1 dependent manner.

Binding with MR1 of antigen-presenting cells (APC) stabilizes the otherwise unstable riboflavin intermediates, which facilitates their transport to the TCR receptor on the surface of MAIT cells to activate antimicrobial responses. Loading of the soluble ligands onto the MR1 molecule has been reported to occur mainly in the endoplasmic reticulum and cell surface [11,12]. The MR1-restriction of MAIT cell activation was further confirmed by demonstrating a lower susceptibility of MR1-deficient mice to bacterial infections as well as the loss of MAIT cell-stimulating ability of bacteria lacking riboflavin biosynthesis enzymes [13,14]. In fact, this mechanism was also demonstrated by a strict correlation between the capacity of bacteria to synthesize riboflavin from guanosine triphosphate (GTP) and the respective level of MAIT cell-activating ability [14]. During an effort to identify important characteristics of the microbe derived MR1 ligands, key insights for the MR1 dependency of MAIT cell activation were gained from Jurkat MAIT cell experimental models. These experiments demonstrated upregulation of cluster of differentiation (CD69) during coincubation of Jurkat cells with *Salmonella typhimurium* infected C1R cells expressing endogenous levels of MR1 [15]. Superinfection then resulted in surface overexpression of MR1 in these cells. More interestingly, co-culture with filtered *S. typhimurium* culture supernatant also resulted in MAIT cell activation indicating that the antigens are secreted and are soluble.

MAIT cells function in two mechanisms. Their ability to recognize microbial antigens presented by APC makes them play important roles in defending against diverse bacterial infections which vary in their capacity of producing the above ligands. Additionally, because of their innate-like nature, they are activated by the proinflammatory cytokines interleukin (IL)-12 and IL-18 in a TCR-independent manner [16]. Upon activation, MAIT cells are involved in immune responses in various ways, including expression of activation markers CD69 and CD25, secretion of proinflammatory cytokines such as interferon-gamma (IFN-γ), tumor necrosis factor-alpha (TNF-α), and IL-17 as well as exerting cytotoxic properties via granzyme B and perforin secretion [17,18,19]. When they were first discovered, MAIT cells were localized in the lamina propria of the intestinal mucosa in relative abundance [7]. Currently, it is well known that MAIT cells are significantly frequent in the colon, peripheral blood, lungs, liver, pancreas, and lymph nodes of humans as well as mice [20,21,22,23]. In humans, they are particularly enriched in the liver where they account for up to 50% of the total resident T cell population [21]. They also represent up to 10% of the total T cells in blood [24]. Apart from their protective effects through antimicrobial responses, MAIT cells have been implicated in chronic pathological mechanisms. They have been reported to be involved in the development and progression of noninfectious diseases, mainly inflammatory, autoimmune, and metabolic pathologies. Considering their frequencies in the body, it can be suggested that MAIT cells can play important roles in several metabolic impairments including fatty liver disease, which is usually associated with changes in the microbial composition of the gut (Figure 1). This review starts by describing the nature of the interaction between precursor compounds of the bacterial riboflavin biosynthesis pathway and MAIT cells. It then discusses the relationship between gut microbiota and MAIT cells in the context of their development and expansion. Finally, it reviews the latest findings of the roles of MAIT cells in different kinds of liver diseases and suggests potential research areas that can be explored to further examine and characterize their exact roles to find clues for therapeutic targetings.

## 2. Microbial Riboflavin Biosynthesis Pathway and MAIT Cell Activation

The exact nature of antigens that activate MAIT cells through the MR1 dependent pathway has not been clearly identified until recently. Several studies have now identified and demonstrated precursors of microbial riboflavin biosynthesis pathway as being the major sources of antigens for stimulation of MAIT cells in vitro [9,25,26]. Although human MAIT cells respond to a variety of bacteria, including *Escherichia coli*, *Klebsiella pneumoniae*, *Staphylococcus aureus*, *Mycobacterium tuberculosis*, and *Saccharomyces cerevisiae*, no activation was observed in the case of viruses or certain bacterial strains [27,28]. This indicates that microbes with defects in the riboflavin synthesis pathway have less capacity to activate MAIT cells. The bacterial riboflavin biosynthesis pathway is an enzymatic cascade that starts from GTP and ribulose-5-phosphate molecules to generate the MAIT cell ligand 5-A-RU (Figure 2). The more potent pyrimidine adduct ligands 5-OE-RU and 5-OP-RU are then produced by a nonenzymatic reaction between 5-A-RU and the small molecules glyoxal and methylglyoxal. However, the overall riboflavin metabolism in bacteria involves a more complex array of enzymes where the production of these ligands can be essentially influenced by several other enzymes involved in the entire cascade of the pathway. Therefore, the MAIT cell activating capacity of a species is strictly correlated to its ability to synthesize riboflavin. This has been confirmed in different bacterial species such as *E. coli*, *Lactobacillus lactis,* and *S. typhimurium* [25,29,30].

Moreover, it has been reported that mutations in enzymes essential for 5-A-RU production prevent MAIT cell activation, whereas the enzymes required in later stages of the pathway did not [25]. This shows that differing pathway profiles in different bacterial taxa result in a unique antigen production pattern and subsequent MAIT cell-activating capabilities. An in vitro study further confirmed this by constructing *ribD* and *ribH* mutant *Salmonella* strains where the former was found to be deficient in its ability to stimulate Jurkat MAIT cells while the latter showed no impact [31]. Another study that used *Lactococcus lactis* mutants also tested their ability to activate Jurkat MAIT cells [25]. It showed that the wild type strain activated MAIT cells in the presence of MR1 expressing APC, indicated by the upregulation of CD69. The addition of exogenous riboflavin was able to abrogate activation. On the other hand, although *ribA* deletion mutant strain could not induce Jurkat MAIT cell activation, the riboflavin overproducing mutant activated them even in the presence of exogenous riboflavin. It can also be suggested that several other adducts produced by condensation of 5-A-RU with a broad range of small molecules other than glyoxal or methylglyoxal exist. This then generates distinct variations in the production of ligands from different microbes. For instance, despite the similar 5-A-RU levels in supernatants from *E. coli* and *S. typhimurium*, a higher level of 5-OE-RU was observed in the former [25]. It has been difficult to examine the mechanisms of interactions between the riboflavin pathway and MAIT cells in vivo. However, few recent studies have developed bacterial infection models of mice to evaluate the importance of microbial riboflavin biosynthesis pathway in inducing in vivo MAIT cell activation (Figure 3). Chen et al. investigated MAIT cell responses in C57BL/6 mice infected with riboflavin gene-competent and deficient bacteria and observed distinct patterns [31]. One week after lung infection with *S. typhimurium*, MAIT cells were enriched only in the riboflavin gene-competent mice where they accounted for up to 50% of the total αβ-T cell population. This accumulation was confirmed to be MR1 dependent as no accumulation was observed in MR1 knockout mutants. Therefore, microbial mediated MAIT cell activation and accumulation in vivo are strictly dependent on the presence of a functional riboflavin synthesis pathway in the stimulating microbe and expression of MR1 in the host mice.

## 3. Gut Microbiota and MAIT Cells

The mammalian gut is colonized by an enormous number of microorganisms that are continuously under multifold interactions with the host’s immune system. The multispecies microbial communities, including bacteria, fungi, and viruses play critical roles in the training and development of major components of the host’s innate and adaptive immune system. Of paramount importance is the mucosal immune system of the intestinal barrier, which is under the highest exposure to the external environment. Therefore, it is under the crucial task of co-existing with a diverse and dynamic population of microbes while protecting the host against pathogens. Under normal conditions termed “eubiosis”, which is mainly characterized by interspecies balance, interactions between the gut barrier and gut microbiota are homeostatic and tightly controlled by both innate and adaptive immune responses [34]. However, this balanced state can be interrupted by changes in the microbiome environment such as diet and alcohol or impaired chronic pathophysiological conditions in the host. This results in “dysbiosis” and alterations in the complex host-microbiota interactions. Particularly, dysbiosis is a key driver for susceptibility to pathogens, aberrant immune responses, and chronic gastrointestinal and cardiovascular diseases [35,36,37]. Each of the different communicable and non-communicable diseases resulting from alteration of the gut microbiota has microbial signatures triggering further pathophysiological progressions in different parts of the host [38].

Members of the gut microbiota are required for the development and stimulation of specific and nonspecific immune system starting from birth and throughout the entire life. Each specific stimulation educates the immune system to learn and differentiate between commensal and pathogenic microbes [39,40]. The interaction between gut microbiota and the host immune system is mainly mediated by microbial molecular signatures called microbe-associated molecular patterns (MAMPs). For instance, receptors known as toll-like receptors (TLR) on the epithelial and lymphoid cells of the intestinal epithelium, recognize a variety of MAMPs and play key roles in the development of the mucosal immune system and differentiation of microbes for responses [41,42]. TLRs differentiate between microbes by either stimulating immune response or promoting immune tolerance based on the type of recognized PAMPs. During the first weeks after birth, TLRs are believed to be less active so that stable colonization takes place. This is a critical stage for the development of the mucosal and systemic immune system. T cells residing in the thymus then develop into different subsets upon facing the respective microbial ligands originating from the gut.

MAIT cells first develop as naïve T cells in the thymus before acquiring memory characteristics and expanding in other tissues including the periphery, mucosa, and liver [20,43]. The development of MAIT cells in the thymus passes through different stages that are defined by the expression of specific markers [44]. In humans, immature MAIT cells are CD161^-^ and are predominant in the thymus with only minor subsets in the circulation. On the other hand, mature MAIT cells express higher levels of CD161 and are significantly populated in tissues. To understand the mechanism by which they expand, researchers employed germ-free mice to localize MAIT cells. Immature MAIT cells were present in the thymus of germ-free mice whereas no mature ones were detected in other tissues. These studies later confirmed that gut bacteria capable of producing ligands for MR1 are required for MAIT cell maturation. On the other hand, MAIT cells do not proliferate in mice colonized by bacteria lacking riboflavin biosynthesis and are therefore unable to trigger MR1 [13]. This indicates that ligand producing bacterial taxa in gut microbiota are the key players in MAIT cell development and activation. Regarding populations, MAIT cells account for only below 0.1% of T cells at birth whereas they represent up to 6−10 and 20−50% of all T cells in blood and liver, respectively in adults [45,46]. Therefore, early colonization of the gut greatly influences the development and characteristics of tissue-resident MAIT cells.

Although the interaction of MAIT cells with the diverse gut microbiota is still unexamined, some studies have investigated physiological responses to microbial communities. Such responses have been found to greatly influence the composition and diversity of the gut microbiota especially in altered environments including dietary stress (Figure 4). MAIT cells have been implicated in inducing dysbiosis during a high-fat diet and viral infections [33,47]. It can also be hypothesized that variable levels of MAIT cell development and activation are initiated by encounters with different types of gut microbes or gut environment. The most dominant microbial phyla in the gut of healthy human subjects are Firmicutes, Bacteroidetes, Actinobacteria, Proteobacteria, Fusobacteria, and Verrucomicrobia, with Firmicutes and Bacteroidetes alone accounting for 90% of the whole gut microbiota population [48]. In a study that analyzed the contribution of individual phyla to MAIT cell activation, Bacteroides, Proteobacteria, Actinobacteria, and Firmicutes activated MAIT cells in decreasing order [14]. Particularly, bacterial species from Bacteroides have been found to be the strongest stimulators of MAIT cells and are usually associated with a higher frequency of circulating MAIT cells. In addition to being an essential part of the gut microflora, they have been detected in breast milk which indicates that it could be one mechanism of how they are established in the infant’s gut and play major roles in MAIT cell development and expansion since early in life [49]. In another similar study that analyzed individual strains of species commonly found in the gut, the MAIT cell activation potential was found to positively correlate with the capacity of riboflavin secretion into the media [50]. *Bacteroides thetaiotaomicron,* which secreted the highest amount of riboflavin, resulted in the highest MAIT cell activation followed by *E.coli* and *Lactobacillus plantarum* which also showed intermediate and lower secretions, respectively. Based on this, it can be concluded that MAIT cells rely on signals from the gut for their differentiation and function through the MR1 dependent pathway.

Considering its pivotal impact on host metabolism and immune homeostasis, alterations in the healthy functioning gut microbiota can result in several inflammatory and metabolic diseases [51,52,53]. Evidence on the direct involvement of MAIT cell activation during pathological progressions caused by dysbiosis is not yet well established. However, recent studies have reported that high-fat diet (HFD) and alcohol-induced dysbiosis disrupt host-microbe interactions partly via the metabolic pathways [53,54]. During dysbiosis, the intestinal barrier is compromised and MAIT cells in the mucosa, mainly the lamina propria, are one of the first immune cells exposed to gut microbes. Moreover, MAIT cell-activating bacteria and microbe derived metabolites can be translocated to other tissues, such as the liver. A recent in vivo study found altered MAIT cell frequency patterns in different tissues and demonstrated a significant reduction in peripheral blood, epididymal adipose tissue, and the ileum of HFD fed murine models [33]. During HFD induced dysbiosis, MAIT cells promoted inflammation that led to impaired glucose and lipid metabolism, providing clear evidence for the direct involvement in metabolic dysfunction and associated diseases. To elucidate the actual mechanisms by which HFD-induced dysbiosis impacts MAIT cell functions, studies have explored the changes in relative abundances of the enzymes of the riboflavin biosynthesis pathway. As a result, distinct profiles in relative abundances of cecal/fecal *ribB, ribH, ribE, ribD, ribBA, ribF,* and *ribA* were observed in mice fed HFD compared to those fed a normal diet (ND) [33]. This indicates that HFD-induced dysbiosis may result in enhanced production or depletion of MAIT cell-activating ligands in the gut depending on other conditions thereby having different endpoint effects on host pathophysiology.

The pathologic phenotypes of MAIT cells during a breach of intestinal barrier caused by dysbiosis has also been shown by a switch to *granzyme B* production and increased production of *IFN-γ* and *TNF-α*, promoting inflammation [55]. On the other hand, MAIT cells were also implicated for impacting the gut microbiota during a high-fat diet. In MR1 deficient (Lacking MAIT cells) HFD mice models, a relative abundance of bacterial taxa such as Actinobacteria and Coriobacteriaceae significantly decreased while those of Clostridiaceae and Lachnospiraceae increased compared to their littermate controls. On the contrary, a significant increase in Actinobacteria and Coriobacteriaceae was observed in TCR Vα19 transgenic (high frequency of MAIT cells) mice of the same diet model [33]. Therefore, gut microbiota and MAIT cells are corresponsive and are suggested to be important targets in the quest to understand the interaction among dysbiosis, the immune system, and disease.

## 4. MAIT Cells and Diseases

Apart from their indispensable roles in the adaptive and innate immune system, T cells are also major drivers of several pathological mechanisms mainly inflammatory and autoimmune abnormalities. Current research on the relationships between T cells and diseases mainly focuses on the well-known CD4^+^ and CD8^+^ T cells which recognize the usual peptide and lipoprotein-based antigens presented by MHC- class I and II antigen-presenting molecules. Unlike conventional T cells, which have diverse TCR repertoires allowing them to fine-tune antigen recognition, MAIT cells have a highly conserved TCR invariant that recognizes only certain microbial metabolites presented by MR1. MAIT cells are most abundant in parts of the body that are closer to the host environment interaction such as the intestinal lamina propria and liver. Therefore, they are continuously exposed to microbial antigens and can be activated in both the MR1 dependent and independent manners. Recently, MAIT cells have started to attract increasing attention from the biomedical research community because of several factors. These include MAIT cell abundance in important tissues such as peripheral blood and the liver, discovery of their target microbes, and the development of advanced tools and techniques to accurately characterize the cells in human subjects, animal models, and in vitro cell cultures. Although MAIT cells have protective roles against several acute bacterial and viral infections, they also have pathogenic roles of sustaining inflammation and cytotoxicity in chronic pathological situations, including chronic viral infections and autoimmune, inflammatory, and metabolic diseases [56,57,58]. A recent study also found that MAIT cells are engaged in the immune response against severe acute respiratory syndrome coronavirus 2 (SARS-CoV-2) and suggested their possible involvement in coronavirus disease 2019 (COVID-19) immunopathogenesis [59]. In another study that applied the *Helicobacter pylori-*infected mice model, reduced gastritis was observed in Mr1^−/−^ mice compared to wild type indicating the deteriorative effect of MAIT cells [60]. Strengthening this, disease severity was increased in mice models with a higher frequency of MAIT cells induced by either bacterial infection such as *Salmonella* or TCR V19α transgenic models [31]. Considering these ambivalent effects of MAIT cells depending on pathological situations, it is necessary to be cautious when attributing MAIT cell activation with protective or deleterious effects before designating them friends or foes.

### 4.1. MAIT Cells and Liver Disease

The liver is the next firewall against enteric commensals when the gut barrier is breached due to inflammation. It is an important organ with extensive involvement in the immune system to maintain homeostasis. Its portal vein, which receives the majority of liver blood supply from the intestinal vessels, plays unique roles in the immune system including resistance to pathogens. The blood entering the liver passes through immune cells in the hepatic sinusoid, and subsequent immune responses are mediated, resulting in the first line of microbial defense. The extensive immune cell population of Kupffer cells, dendritic cells, natural-killer (NK) cells, and T cells including MAIT cells in the liver are responsible for such functions. With continuous exposures and chronic pathological progressions, resident liver T lymphocytes are considered to play major roles. Because of the high population of MAIT cells in the liver (constituting up to 50% of total T cells), it is evident that they have an important impact on the regulation of several kinds of liver pathophysiology (Table 1). A study that characterized their localizations and phenotypes in the liver during chronic liver injuries found that MAIT cells are mainly concentrated in portal tracts and hepatic sinusoids [61]. Similar studies also have demonstrated more prevalence of MAIT cells in portal tracts than in the main parenchymal tissue [62]. Their tissue homing is usually reflected in their phenotypes. Several studies have demonstrated their involvement in liver fibrosis, a condition that commonly occurs in response to chronic liver injury [63,64,65]. Their ability of tissue homing is attributed to their tissue homing receptors C-C chemokine receptor (CCR) 5 and CCR6 enabling them to migrate to inflammation sites in the liver [21].

MAIT cells in the liver highly express CD25, CD69, and CD38 when activated. This is believed to be stimulated by the translocation of microbes or microbial products from the gut. Additionally, liver MAIT cells are activated by IL-12 and IL-18 which are mainly produced by Kupffer cells after viral infection. A previous study described the licensing of TCR-mediated liver MAIT cell stimulation by IL-17 that resulted in the production of large quantities of IFN-γ [46]. This indicates that MAIT cells can be considered as key modulators of inflammation and microbial defense in the liver resulting in both protective and detrimental effects depending on the pathological conditions.

#### 4.1.1. Nonalcoholic Fatty Liver Disease (NAFLD)

Nonalcoholic fatty liver disease (NAFLD) is a leading cause of chronic liver disease and one of the major public health problems in the world [77,78]. Diets high in fat are the main risk factors for NAFLD through inducing higher expression of lipogenesis genes, elevated production of proinflammatory cytokines and reactive oxygen species as well as alteration of the gut microbiome and intestinal barrier [79]. Although a direct role of MAIT cells in the progression of NAFLD has not been recorded, there are several pieces of evidence showing their involvement in the context of alteration of their frequencies and phenotypes, alteration of gut microbiota, disruption of the intestinal barrier, and overgrowth of bacterial pathogens. A recent study that analyzed peripheral blood mononuclear cell (PBMC) of NAFLD patients revealed a decreased MAIT cell frequency compared to healthy controls [32]. This same study found increased MAIT cell frequency in the liver and elevated surface expression of MR1 in Kupffer cells of NAFLD patients. This shows that stimulation of APCs results in the trafficking of MR1 from the endoplasmic reticulum to the surface to present antigens to MAIT cells [80]. Methionine and choline-deficient diet (MCD), which is commonly applied to induce nonalcoholic steatohepatitis (NASH) resulted in a more severe hepatic steatosis in MR1 knockout mice compared to wild type. Another study also reported that MR1 knockout mice had a higher bacterial burden following intraperitoneal injection of *E. coli* or intravenous injection of *Mycobacterium abscessus* [13]. Therefore, the link between MAIT cells and lipid metabolism is a promising research area in the context of dysbiosis.

#### 4.1.2. Alcoholic Fatty Liver Disease (ALD)

At the beginning stage, alcoholic liver disease (ALD) is primarily characterized by liver inflammation, hepatocyte regeneration disorder, and bacterial infection [81]. The higher infection rate observed in ALD patients is attributed to alteration of gut microbiota, bacterial overgrowth, and translocation to the portal vein [82,83]. In a study that explored the roles of MAIT cells in severe alcoholic hepatitis (SAH), a significant reduction of MAIT cells was observed in the circulation [84]. The depletion was suggested to be caused by their migration to the liver. Additionally, the remaining population of MAIT cells in ALD patients exhibit altered phenotypes characterized by overexpression of activation markers including CD69. However, the phenotypically hyperactivated MAIT cells were found to be functionally deficient. This was demonstrated by measuring and comparing the antibacterial cytokine production of MAIT cells from ALD patients versus healthy controls after *E. coli* stimulation. Although MAIT cells from both sources were able to produce similar levels of IFN-γ and TNF-α, only cells from healthy controls were able to produce IL-17. Moreover, the depleted MAIT cell frequency in ALD patients was confirmed to have been induced by fecal bacteria. Another study that investigated MAIT cells in alcoholic hepatitis (AH) patients reported an elevated prevalence of soluble factors responsible for MAIT cell activation [66]. AH is mainly characterized by impaired integrity of the gastro intestine (GI) that leads to increased translocation of bacterial components to the liver. On the other hand, chronic AH is usually accompanied by microbial infections that can result in cytokine-mediated MAIT cell activation. The elevated expressions of IL-12R and IL-18R by MAIT cells during infection allows for their TCR-independent stimulation and increased expression of IFN-γ [16].

#### 4.1.3. Viral Hepatitis

Chronic hepatitis B and C viruses are the most important causes of liver diseases [85]. Whereas the major function of MAIT cells is inhibition of bacteria through the MR1-dependent pathway, it can also play roles in various types of liver diseases including viral hepatitis [86]. This is particularly interesting in the context of the importance of IFN-γ, an important antiviral cytokine, and interaction of antigen-presenting cells during viral infection. A study by Bolte et al. reported that inflammatory cytokines resulted in activation-induced MAIT cell death in viral hepatitis patients. As a result, significant depletion of MAIT cell frequency is observed in the liver of chronic hepatitis C virus (HCV) infection patients. The frequency of MAIT cells in the liver was found to be inversely proportional with inflammation and fibrosis, suggesting the involvement of MAIT cells in hepatic inflammation during viral hepatitis infection [70]. Here, proinflammatory monocytes are suggested to play important roles in the activation of MAIT cells during HCV infection. In addition to their function of presenting bacterial riboflavin metabolite antigens to induce MR1-dependent MAIT cell activation, they also release the cytokines IL-12 and IL-18 in response to HCV [87,88]. In the case of liver injury caused by chronic hepatitis B virus (HBV) infection, the cytotoxic function of MAIT cells was demonstrated to be reduced, which is mainly attributed to the decreased expression of the early MAIT cell activation marker CD69 [73]. In summary, these results show that MAIT cells have larger roles in liver pathogenesis in addition to the MR1 dependent activation pathway.

## 5. Concluding Remarks and Future Perspectives

MAIT cells are now considered to be an important part of the mammalian immune system, representing the majority of the T cell population. Although the knowledge about MAIT cells is rapidly accumulating, understanding their roles in various diseases is still at the beginning stage. Most of the studies in the context of various types of liver (an organ where they are dominant) diseases demonstrated depletion of MAIT cells in the blood with a relative tendency to accumulate in the liver. However, it is impossible to reach a unified conclusion on their exact roles and mechanisms and their dynamics with changes in other pathological factors. Given that MAIT cell activation occurs in the TCR-mediated and cytokine-mediated ways that are independent of each other, which one dominates in a particular liver injury condition should be an interesting research target. Of paramount importance is their responses according to changes in the gut microbiota which is in direct communication with the liver. It is now clear that the most characterized stimulating microbial ligands are the precursors of the riboflavin biosynthesis pathway. These metabolites are produced by an extensive number of members of the gut microbiota. It is apparent that MAIT cells as pathogen defenders do not act up on every microflora that is capable of synthesizing riboflavin. Here, a huge gap exists in the knowledge about how they discriminate between the beneficial and pathogenic taxa. This is an interesting area that should be explored. The antigen-presenting molecule MR1 has been found to have sufficient plasticity to bind to a wide range of chemical entities other than those identified and characterized so far [26]. Therefore, further studies are required to identify and map other classes of potential MR1 binding ligands. Additional characterization is also needed to determine whether such binding components activate or inhibit MAIT cells. The gut-liver axis is not only an important factor in the TCR-mediated but also the cytokine-mediated MAIT cell activation. The antigen-presenting cells in the liver including B cells, dendritic cells, T cells, and macrophages mostly function in response to microbial antigens from the gut to release IL-12 and IL-18. Therefore, activation of MAIT cells is directly dependent on the microbial population and diversity of the gut. Based on this, establishing systematic patterns of actual contributions of individual microbial communities of the gut on MAIT cell phenotype dynamics might help in mapping and correlating observed microbial signatures (metagenomic or metabolomic) and specific pathophysiological implications. This can then be used as a good tool in the design of therapeutic interruptions specific to different conditions as various taxonomic groups might trigger different phenotypes depending on different types of pathological conditions. Regarding the already identified and characterized MR1 ligands, a systematic method can be applied to monitor their relative abundance. Ongoing studies are promising to develop tools to determine the relative abundance of the metabolites directly or predict them indirectly from metagenomic and metabolomic data pools. Gene copies for enzyme orthologs involved in the production of such metabolites can be analyzed and help predict and estimate the regulation of the riboflavin biosynthesis pathway. Moreover, the fact that several other unidentified adducts are produced by condensation of 5-A-RU with a broad range of small molecules other than glyoxal or methylglyoxal may generate distinct variations in the production of ligands that activate MAIT cells. In conclusion, further insights into MAIT cells can advance the quest for designing biomarkers for diseases and therapeutic approaches.

## Figures and Tables

**Figure 1 microorganisms-09-00296-f001:**
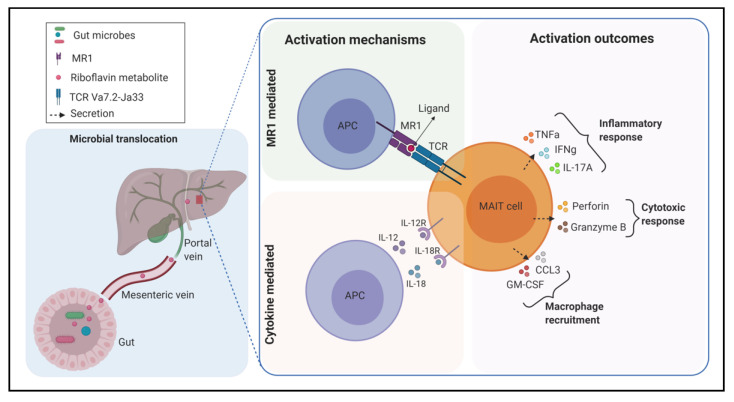
A schematic representation of the mechanisms of mucosal-associated invariant T (MAIT) cell activation in the liver. MAIT cell-stimulating members of the gut microbiota or microbial ligands are translocated to the liver through the portal vein. In the T-cell receptor (TCR) mediated manner, antigen-presenting cells (APC) present riboflavin metabolite ligand loaded on major histocompatibility complex (MHC) class I- related (MR1) molecule to MAIT cells. In the cytokine-mediated manner, APCs release the cytokines interleukin (IL)-12 and IL-18 to activate MAIT cells via their receptors. Upon activation, MAIT cells exert different kinds of immune responses. TNF-α, tumor necrosis factor-alpha; IFN-γ, interferon-gamma; IL-12R, IL-12 receptor; CCL3, CC-chemokine ligand 3; GM-CSF, granulocyte-macrophage colony-stimulating factor. Illustration created with https://biorender.com/.

**Figure 2 microorganisms-09-00296-f002:**
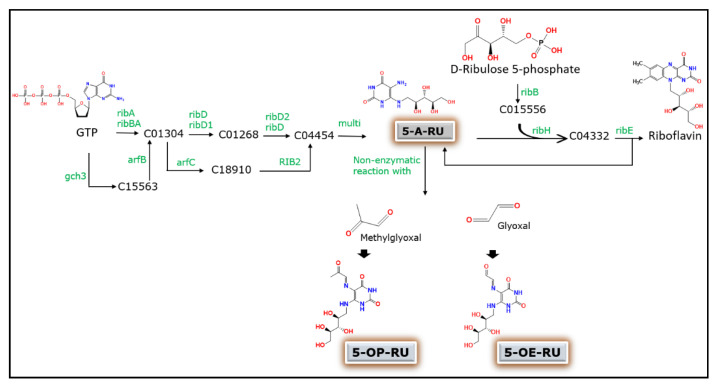
Schematic representation of the generation of MAIT cell-activating ligands during microbial riboflavin biosynthesis pathway. GTP, guanosine triphosphate; 5-A-RU, 5-Amino-6-D-Ribitylaminouracil; 5-OE-RU, 5-(2-oxoethylideneamino)-6-D-ribitylaminouracil; 5-OP-RU, 5-(2-oxopropylideneamino)-6-D-ribitylaminouracil; C number prefixes are compound entries generated as intermediates during the biosynthesis pathway adopted from Kyoto Encyclopedia of Genes and Genomes (KEGG) database (https://www.genome.jp/kegg-bin/show_pathway?map00740); rib, KEGG enzyme entries involved in each step of the pathway.

**Figure 3 microorganisms-09-00296-f003:**
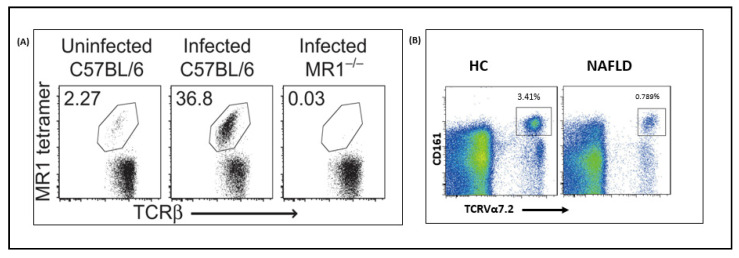
A cross-talk between MAIT cell expansion and bacterial ligands. (**A**) Flow cytometry scatter plots showing TCRβ+ lymphocytes (with a percentage of MAIT cells) harvested from mice lung tissue in a bacterial infection model, (**B**) flow cytometry scatter plots from the blood of healthy and fatty liver disease patients. MR1, major histocompatibility complex class I-related; HC, Healthy control; NAFLD, Nonalcoholic fatty liver disease; CD, Cluster differentiation; TCR, T-cell receptor. Figures reprinted from [31,32,33].

**Figure 4 microorganisms-09-00296-f004:**
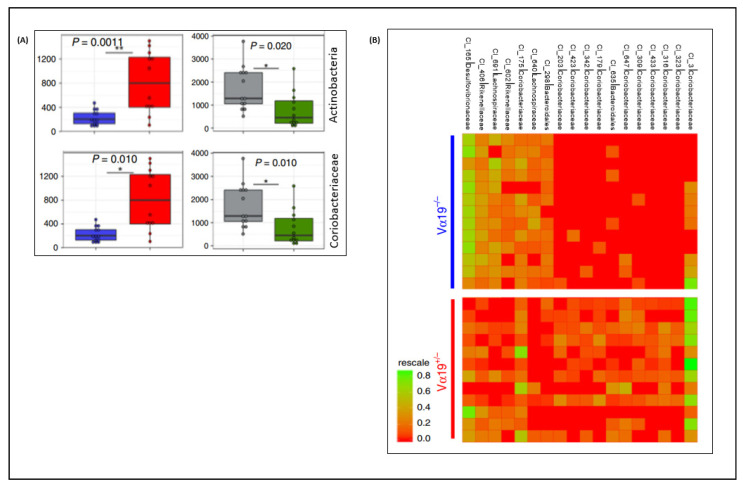
Impact of MAIT cells on the gut microbiome. (**A**) significant differences in relative abundances of Actinobacteria and Coriobacteriaceae between high and lower MAIT cell frequency mice lines during a high-fat diet (Blue Vα19^−/−^; Red, Vα19^+/−^; gray, MR1^+/−^, green, MR1^−/−^); (**B**) Heatmap showing significant differences between OTU clusters of the Vα19^−/−^ and Vα19^+/−^ mice microbiota samples. Figures reprinted from [33,47].* *p* < 0.05, ** *p* < 0.01.

**Table 1 microorganisms-09-00296-t001:** Roles of MAIT cells in various liver pathogenesis conditions.

Disease	Study Models	Frequency and Phenotype of MAIT Cells	Remarks	Reference
NAFLD	Mr1 KO mice model -NAFLD Patients	Frequency: Blood↓, Liver↑ Markers: Blood CD69, CD38, HLA-DR, PD-1, CXCR6, CCR5↑; Liver MR1 (Kupffer cells) ↑	Deficiency triggers steatohepatitis	[32]
ALD	AH patients	Frequency: Blood↓, Liver↑ Markers: CD161↑, CD69↑, CD38↑, HLA-DR↑, Granzyme B↑ Activation factors ↑	Alcohol cessation reverses MAIT cell abnormalities	[66]
Cirrhosis	-Cirrhotic ALD patients -CCl4-induced liver injury mice model	Frequency: Blood↓, Liver↑ Markers: Blood CD25↑, CD69↑, IL-17↑, Granzyme B↑, IFN-γ↑, TNF↑; Liver: IL-17 ↑, MR1 ↑	-Profibrogenic -Macrophage recruitment	[63]
HCC	HCC patients	Frequency: Blood↓, Tumor ↓ Markers: Tumor CD38↑, HLA-DR↑, CCR6↓, CXCR6↓, CCR9↓ Blood CD45RO; Liver CD28↓, CD127↓	Infiltration of tumor by MAIT results in dismal clinical outcomes	[67,68,69]
HCV	Chronic HCV infection	Frequency: Blood↓, Liver↓ Markers: Granzyme B↓, IFN-γ↓, CD69↓, IL-12↑, IL-18↑,	Inverse correlation between MAIT cell frequency and liver inflammation	[70,71]
HBV	Chronic HBV infection	Frequency: Blood↓, Liver↓ Markers: Granzyme B↓, IFN- γ↓, CD69↓,	Impaired cytotoxic function with reduced CD69 expression	[72,73]
AILD	AILD patients	Frequency: Blood↓, Liver↓ Markers: CD38↓, HLA-DR↓, CTLA-4↓, IFN-γ↓, Foxp3↑, CD8↑, CD161↑, Granzyme B↑	Promote Profibrogenic Hepatic Stellate Cell Activation	[56,74]
IBD	IBD patients	Frequency: Blood↓, Inflamed tissue↑ Markers: IFN-γ↓, IL-17↑, CD69↑, PD-1↑, annexin V↑, CCL20↑, CXCL10↑, CXCL16↑, CCL25↑	MAIT cells migrate to inflamed tissue	[75,76]

↑ increased frequency/expression level; ↓ decreased frequency/expression level; NAFLD. nonalcoholic fatty liver disease; ALD, alcoholic liver disease; AH, alcoholic hepatitis; HCC, hepatocellular carcinoma; HCV, hepatitis C virus; HBV, hepatitis B virus; AILD, Autoimmune liver disease; IBD, inflammatory bowel disease; MR1, MHC class I-related molecule; KO, knock out; MAIT, mucosal-associated invariant T; CD, cluster of differentiation; PD, programmed cell death protein; CCR, C-C chemokine receptor; CXCR, C-X-C Motif Chemokine Receptor; IL, interleukin; IFN, interferon; TNF, tumor necrosis factor; Foxp3, forkhead box P3; CCL, Chemokine (C-C motif) ligand.

## Data Availability

Data is contained within the article.

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
