# Peer review of "Pathophysiological Roles of Mucosal-Associated Invariant T Cells in the Context of Gut Microbiota-Liver Axis"

_microorganisms, 2021, doi:10.3390/microorganisms9020296_

Round 1

Author Response

SN

Reviewer’s Comments and suggestions

Author’s responses

1

The review Pathophysiological Roles of Mucosal-Associated Invariant T Cells in the Context of Gut Microbiota-Liver Axis is a well readable treatment of the current state of the art of knowledge about Mucosal-Associated Invariant T (MAIT) cells. The strength of the manuscript is that it describes first the general knowledge about the biology of MAIT cells including their activation in two different and independent ways through 1) MHC calls Irelated molecules, for which bacterial produced riboflavins acts a ligands and 2) through binding of cytokines. A second strength is that it is not only a generic MAIT cell review, but it takes the knowledge about activation to specifically outline the involvement in liver related diseases. It becomes clear that MAIT cells play a central role in a range of liver diseases however there are still gaps in the knowledge, which are addressed in the final concluding remarks section.

ü  We are grateful for your detailed review, positive comments and complements as well.

2

I have a few major and some minor comments outlined below. The major comments relate to the figures and some of the displayed results that could be improved.

2.1. Figure 3A could be improved by adding a cluster tree on both axis, at least on the x-axis in order to show that normal diet and high fat diet provide distinct profiles for the abundance of the abundance of different riboflavin biosynthesis genes.

ü  We do agree with the reviewer’s point of view that heatmap with cluster tree would have visualized the patterns better. Unfortunately, we have difficulty retrieving the original raw data at this moment and we are unable to build a heat map with cluster tree. This said, we believe the distinct profiles in the two groups are clearly noticeable (All ND clustering left and all HFD clustering right) and the relationships between individual samples can still be recognized with a closer look. We hope the reviewer understands the limitation.

2.2. Figure 4B this correlation 1) Seems not be very strong, 2) seems to refer to a Pearson correlation which is valid for normally distributed data (the impression is that at least the abundance of Bacteroides spp. Is not normally distributed, and hence a Spearman rank correlation is more suited), 3) several different phenotypic states are mixed up (healthy controls, diseased patients) and obviously, the correlation seems to be valid for healthy patients, but not for the diseased. I suggest to do it properly or remove completely.

ü  We understand there is no consistent correlation in both the healthy control and viral hepatitis samples. However, the purpose of including this figure was only general in a way that MAIT cell frequency may reflect differences in gut microbiota composition in the context of relative abundance of a particular taxon or vice versa. Therefore, we believe it can provide a bit of information regarding the described point of view. To elaborate and strengthen this concept, we have added the following content from line 251-256 with proper reference.

“Particularly, bacterial species from Bacteroides have been found to be strongest stimulators of MAIT cells and are usually associated with higher frequency of circulating MAIT cells. In addition to being essential part of the gut microflora, they have been detected in breast milk which indicates that it could be one mechanism how they are established in infant gut and play major roles in MAIT cell development and expansion since early in life’

2.3. Figure 4C as suggested before, the heatmap could be improved by adding cluster trees to visualize the relationship of individuals samples and the bacterial OTUs

ü  To improve the heatmap, we have added each OTU on the x-axis and made revisions in the legend for Figure 4C at lines 272-273 in the revised version. However, like our response for the suggestion above, we agree that cluster tree could have presented the results better, but we do not have the raw data in hand. We would like to remind the reviewer that this heatmap mainly attempted to convey information about the distinct profiles of OUT of mice with differences in MAIT cell frequencies.

2.4. For better readability I suggest to have a spacing line between Figure legends and subsequent main text.

ü  According to the reviewer’s suggestion, spacings have been added at lines 122, 149, 186, 258 and 343 in the revised version.

2.5. L166 in vivo should be in italics

ü  The font is changed to italic at line 169 of the revised version

2.6. L248 (figure legend) MAIR cell should be MAIT cell, however, the whole sentence should be re-phrased: e.g.: … between high and lower MAIT cell frequency in high fat diet-fed mice models.

ü  In agreement with the reviewer, the sentence from line 251-252 is revised as follows.

‘significant differences in relative abundances of Actinobacteria and Coriobacteriaceae between high and lower MAIT cell frequency mice lines during high fat diet’

2.7. L270 remove additional space in front of IFN

ü  The extra space before IFN is removed at line 291

2.8. L273/275/276 start bacterial taxa names with capital letter

ü  Corrections are made at lines 294 and 297 of the revised manuscript

2.9. L301 helicobacter pylori, start with capital letter and put in italicsL364 E. coli in italics

ü  Changes are made at lines 324 and 260 of the revised manuscript

2.10. L372 severe

ü  The word is corrected at line 396

2.11. L385 please write out GI and introduce abbreviation in brackets

ü  We have added the full form of the phrase at line 409

2.12. L412 human immune system? You cite many studies utilizing mice, hence it might be an important part of mammalian immune system.

ü  We agree with the reviewer that ‘mammalian’ is more appropriate. We thank the reviewer for noticing and we have changed it at line 436.

2.13. L422 important should be replaced by importance

ü  The word is corrected at line 446.

Additional revisions

Line 15: We have added the abbreviation ‘MR1.

Lines 36-38: More introductory information is added.

Line 41-43: We have added the following content to include more background information.

‘Several studies have explored and described the importance and natures of various types of T cell subsets with effector and memory immune functions in healthy as well as dis-eased individuals.

Lines 70-72: We have added the following content to elaborate mechanism of MR1 ligand loading.

‘Loading of the soluble ligands onto the MR1 molecule has been reported to occur mainly in the endoplasmic reticulum and cell surface.

Lines 138-139: An expression was revised.

Line 146: The word adopted is added.

Line 151: grammar change

Line 156, 157: Font corrections

Lines 218-220: Information is added

‘T cells residing in the thymus then develop into different subsets upon facing the respec-tive microbial ligands originating from the gut.

Lines 302-304: The following content is added.

‘Apart from their indispensable roles in the adaptive and innate immune system, T cells are also major drivers of several pathological mechanisms mainly inflammatory and autoimmune abnormalities’.

Reviewer 2 Report

The review provided by Gebru et al. summarizes the current knowledge on the role of mucosal-associated invariant T (MAIT) cells regarding liver diseases and the gut microbiome. They first give insights in the physiological role and biology of MAIT cells and then discuss their role in different liver diseases like NAFLD or viral hepatitis. The manuscript is clearly written also for people outside the field. The authors also have expertise in this field as shown by recent publications. I only have minor comments:

  • Figure 4C: Is the annotation of the heatmap concerning the different OTU clusters missing?
  • Line 338: is that correct? “highly express both CD25, CD69 and CD38”
  • Line 383+384+386: Regarding the explanation of the abbreviation “alcoholic hepatitis”: Please explain it with the first mentioning in line 383.
  • Minor typing/formatting errors: line 77, 146-147, 160, 166, 248, 250.

Author Response

SN

Reviewer’s Comments and suggestions

Author’s responses

1

The review provided by Gebru et al. summarizes the current knowledge on the role of mucosal-associated invariant T (MAIT) cells regarding liver diseases and the gut microbiome. They first give insights in the physiological role and biology of MAIT cells and then discuss their role in different liver diseases like NAFLD or viral hepatitis. The manuscript is clearly written also for people outside the field. The authors also have expertise in this field as shown by recent publications. I only have minor comments:

ü  We are grateful to the time, efforts, and positive comments of the reviewer.

2

I only have minor comments:

2.1. Figure 4C: Is the annotation of the heatmap concerning the different OTU clusters missing?

ü  We thank the reviewer for noticing this. To improve the heatmap, we have added the OTU clusters on the horizontal axis and reorganized Figure 4C in the revised version of the manuscript.

2.2. Line 338: is that correct? “highly express both CD25, CD69 and CD38”

ü  We are assuming that the reviewer is asking about the grammatical structure of the sentence. We attempted to consider CD25 and CD69 as different group of markers from CD38. To correct it, the sentence at line 362 is revised by removing the word ‘both’. We hope this was exactly the reviewer’s issue.

2.3. Line 383+384+386: Regarding the explanation of the abbreviation “alcoholic hepatitis”: Please explain it with the first mentioning in line 383.

ü  We thank the reviewer for noticing this. We have added the full name and proper abbreviation as well as removed unnecessary texts at lines 407, 408 and 410 of the revised version of the manuscript.

2.4. Minor typing/formatting errors: line 77, 146-147, 160, 166, 248, 250.

ü  The small letter j is converted to capital letter at line 79 and 156, the extra space before semi colon is removed at line 157, the phrase in vivo is reformatted to italic at line 176.

Additional revisions

Line 15: We have added the abbreviation ‘MR1.

Lines 36-38: More introductory information is added.

Line 41-43: We have added the following content to include more background information.

‘Several studies have explored and described the importance and natures of various types of T cell subsets with effector and memory immune functions in healthy as well as dis-eased individuals.

Lines 70-72: We have added the following content to elaborate mechanism of MR1 ligand loading.

‘Loading of the soluble ligands onto the MR1 molecule has been reported to occur mainly in the endoplasmic reticulum and cell surface.

Lines 138-139: An expression was revised.

Line 146: The word adopted is added.

Line 151: grammar change

Line 156, 157: Font corrections

Lines 218-220: Information is added

‘T cells residing in the thymus then develop into different subsets upon facing the respec-tive microbial ligands originating from the gut.

Lines 302-304: The following content is added.

‘Apart from their indispensable roles in the adaptive and innate immune system, T cells are also major drivers of several pathological mechanisms mainly inflammatory and autoimmune abnormalities’.

Round 2

Reviewer 1 Report

Dear authors, 

thanks a lot for the improved version of your review. Most of my remarks and criticism has been addressed very well and contributed to the improvement of the manuscript. 

However, I have to come back to two issues addressed before.

  1. your comment on my suggestion for improvement of figure 3A is quite odd. Retrieving data for a figure you are using in your review should not be the limit of improving the figure. This is extremely disturbing, when you present data but don't know where the data are. Is this really a trustworthy result? Using image processing software you might be able to to convert the figure into data according to the shades of the different colours (probably dependent on the resolution of the image). However, as sources for this figure you also cite the literature (31-33) and you should try to contact all of those authors and check whether one of them is left with a copy of the respective data set. Otherwise, I'm not sure who good it is to present such unreliable data in a review. 
  2. The correlation questioned in figure 4 B can not be used on fusing different phenotypes together. For me it seems that this correlation shows the normal state of healthy individuals and obviously, this correlation breaks down under disease status. Thus, it might be important to report for each status the correlation separately. Might be that there are not sufficient sample sizes available, thus there are two options either do not show it or bring it up in suggestions/conclusions/ way forward that such data have to be generated in a larger scale than what is present now.

Author Response

Dear authors,

thanks a lot for the improved version of your review. Most of my remarks and criticism has been addressed very well and contributed to the improvement of the manuscript.

However, I have to come back to two issues addressed before.

ü  We thank the reviewer for approving most of the first-round revisions and coming back for issues that require further modifications.

1. your comment on my suggestion for improvement of figure 3A is quite odd. Retrieving data for a figure you are using in your review should not be the limit of improving the figure. This is extremely disturbing, when you present data but don't know where the data are. Is this really a trustworthy result? Using image processing software you might be able to to convert the figure into data according to the shades of the different colours (probably dependent on the resolution of the image). However, as sources for this figure you also cite the literature (31-33) and you should try to contact all of those authors and check whether one of them is left with a copy of the respective data set. Otherwise, I'm not sure who good it is to present such unreliable data in a review.

ü  We have made an understanding that this figure presents less reliable data than we thought. Therefore, we have decided to remove it. Instead, we have added a content that describes the importance of the information that was intended to be presented by that figure as follows from line 300-307 on the new version.

‘To elucidate the actual mechanisms by which HFD induced dysbiosis impacts MAIT cell functions, studies have explored the changes in relative abundances of the enzymes of riboflavin biosynthesis pathway. As a result, distinct profiles in relative abundances of cecal/fecal ribB, ribH, ribE, ribD, ribBA, ribF and ribA were observed in mice fed HFD com-pared to those fed normal diet (ND). This indicates that HFD induced dysbiosis may result in enhanced production or depletion of MAIT cell activating ligands in the gut de-pending on other conditions thereby having different endpoint effects on host pathophysiology.’

``

ü  Based on the reviewer’s comment raised again, we have addressed this issue by removing the mentioned figure and added the following content in the ‘Concluding Remarks and Future Perspectives’ section from line 485-491.

‘Based on this, establishing systematic patterns of actual contributions of individual microbial communities of the gut on MAIT cell phenotype dynamics might help in mapping and correlating observed microbial signatures (Metagenomic or metabolomic) and specific pathophysiological implications. This can then be used as a good tool in the design of therapeutic interruptions specific to different conditions as various taxonomic groups might trigger different phenotypes depending on different types of pathological conditions.’

Round 3

Reviewer 1 Report

Thanks for your revised version. I didn't understand quite well why the file named supplementary material just contained the manuscript in track changes mode. Is that a mistake or intentional?

However, my previous concerns have been addressed. Although it would have been better to do the required analyses instead of just removing, but it is fine with me. However, after removal of a section from Fig. 4, please rearrange the remaining parts to delete the large white space.